# Atroposelective hydroarylation of biaryl phosphines directed by phosphorus centres

Zexian Li[1,2,4], Minyan Wang[2,4], Youqing Yang[1], Yong Liang[2], Xiangyang Chen[3], Yue Zhao[2], K. N. Houk[3] & Zhuangzhi Shi[1,2] ✉

Prized for their ability to generate chemical complexity rapidly, catalytic carbon–hydrogen (C–H) activation and functionalization reactions have enabled a paradigm shift in the standard logic of synthetic chemistry. Directing group strategies have been used extensively in C–H activation reactions to control regio- and enantioselectivity with transition metal catalysts. However, current methods rely heavily on coordination with nitrogen and/or oxygen atoms in molecules and have therefore been found to exhibit limited generality in asymmetric syntheses. Here, we report enantioselective C–H activation with unsaturated hydrocarbons directed by phosphorus centres to rapidly construct libraries of axially chiral phosphines through dynamic kinetic resolution. High reactivity and enantioselectivity are derived from modular assembly of an iridium catalyst with an endogenous phosphorus atom and an exogenous chiral phosphorus ligand, as confirmed by detailed experimental and computational studies. This reaction mode significantly expands the pool of enantiomerically enriched functional phosphines, some of which have shown excellent efficiency for asymmetric catalysis.

Generation of enantiopure molecules that operate efficiently with ideal atom and step economy is a long-standing challenge in organic synthesis[1–3]. Catalytic asymmetric C–H activation[4–14] provides a reliable solution to this challenging task. Since a complex molecule typically contains multiple C–H bonds with comparable strengths and steric environments, the most successful method for this transformation is the use of a directing group[15,16], either inherent or preinstalled in organic molecules, to position the metal catalyst at a particular C–H bond in a chiral environment (Fig. 1a)[17–20]. In this context, treatment of chiral ligands with metal catalysts has proven to be extremely effective when directed by aromatic nitrogen heterocycles[21,22], carbonyl groups[23–25], and amino derivatives[26,27]. Compared to oxygen and nitrogen atoms, phosphorus coordinates strongly with transition metals and is therefore challenging to use as a director in catalytic C–H activation[28]. Substantial progress has been made in ligand modification through phosphorus-directed C–H activation[29–41]. We have also

demonstrated the viability of using phosphorus directing groups for the site-selective C–H functionalization of indoles at the benzene core[42,43]. Despite these advances, asymmetric C–H activation directed by a phosphorus center has not yet been overcome.

Chiral biaryl phosphines are a class of promising ligands and organocatalysts and have become tremendously important in modern organic chemistry[44–47]. Preparations of these molecules typically require multistep syntheses, in which chiral auxiliaries and kinetic resolution are used most often in practice[48,49]. Palladium-catalyzed asymmetric C–C[50–52] and C–P[53,54] coupling of aryl (pseudo)halides have been disclosed for the synthesis of axially chiral phosphines. More recently, a series of catalytic asymmetric C–H activation strategies have also been developed to build the biaryl phosphines (Fig. 1b)[55–59]. However, the directing group in these methods is limited to O atom, and additional step for reduction of the formed phosphine oxides is needed. We reasoned that the development of a general strategy to

[1]Key Laboratory of Green and Precise Synthetic Chemistry and Applications, Ministry of Education, Huaibei Normal University, Huaibei 235000, China. [2]State Key Laboratory of Coordination Chemistry, Chemistry and Biomedicine Innovation Center (ChemBIC), School of Chemistry and Chemical Engineering, Nanjing University, Nanjing 210093, China. [3]Department of Chemistry and Biochemistry, University of California, Los Angeles, CA 90095, USA. [4]These authors contributed equally: Zexian Li, Minyan Wang. ✉e-mail: shiz@nju.edu.cn

**Fig. 1 | Background and discovery. a** Pioneering examples of catalytic enantio-selective C−H activation assisted by various directing groups. **b** State-of-the-art methods for catalytic asymmetric synthesis of chiral biaryl phosphines by O-directed C−H activation. **c** Enantioselective P(III)-directed C−H activation enabled by chiral phosphorus ligands.

enantioenriched phosphorus ligands through one-step syntheses would likely have a broad impact on asymmetric catalysis. Herein, we report that libraries of chiral biaryl phosphines can be generated by C−H activation with high regio-, stereo- and enantioselectivity (Fig. 1c). Chirality transfer occurs by combining a metal catalyst with a phosphorus directing group and a chiral phosphorus ligand. The main challenges for this process include strong background reactions with stoichiometric phosphines leading to racemization and formation of a thermodynamically stable metal-ligand complex resulting in low conversion. Therefore, the metal catalyst must distinguish between these two different phosphorus atoms and form a high equilibrium population of the required intermediates to achieve good outcome.

## Results

### Reaction design

We started our project by identifying a class of biaryl phosphines for asymmetric C−H activation with unsaturated hydrocarbons, which would enable access to atropisomers through dynamic kinetic resolution[60,61]. Phosphine **1a** fulfils this criterion and exhibits a barrier of 22.0 kcal/mol for atropisomer interconversion and a sufficiently high rotational barrier for formation of olefination product **3aa** (~45 kcal/mol) from alkyne **2a** (Table 1). When the reaction was conducted with [Ir(cod)Cl]$_2$ (5.0 mol%) as a catalyst and chiral diene **L1**[62] as a ligand in toluene at 70 °C under a N$_2$ atmosphere, the desired product **3aa** was formed in racemic with 33% yield (entry 1). The reaction was then investigated by using the iridium catalyst with a variety of chiral ligands to obtain the enantioselectivity. The use of Carreira ligand **L2** led to product **3aa** in 79% yield with 51% ee (entry 3). Changing the ligand to the chiral spiro phosphoramidite **L3** provided superior results with alkyne **2a**, delivering product **3aa** in 82% yield and 97% ee (entry 3). Here the absolute stereochemistry of product **3aa** was determined by X-ray crystallographic analysis. Loading the BINOL-

derived phosphoramidite **L3** bearing an NMe$_2$ motif decreased the enantioselectivity to 66% (entry 4). However, treatment of a TADDOL-derived ligand **L5** in the system became very sluggish, leading to product **3aa** only in trace amounts (entry 5). Other solvents, such as THF and DCM gave much lower enantioselectivities and yields (entries 6-7). Conducting the reaction at 90 °C further improved the yield of **3aa** to 88% yield but with a reduced enantioselectivity (entry 8). Low efficiency was observed by conducting the reaction at room temperature (entry 9). It is noteworthy that other iridium sources like [Ir(coe)$_2$Cl]$_2$ also maintained an acceptable enantioselectivity (entry 10), but other transition metals such as [Rh(cod)Cl]$_2$ gave a remarkably reduced ee value (entry 11) and Pd(OAc)$_2$ completely failed (entry 12). Decreasing the [Ir(cod)Cl]$_2$ loading to 2.5 mol% led to a substantial decrease in the yield (entry 13), and a control experiment revealed that the reaction did not proceed without the metal catalyst (entry 14).

### Scope of the methodology

With the optimal reaction conditions in hand, we evaluated the scope of two components in this reaction (Fig. 2). The scope of phosphines **1** was first examined with alkyne **2a**. Phosphines with electron-donating groups, including methyl (**1b-e**), phenyl (**1f**) and ether (**1g-i**) substituents at the P-phenyl ring or naphthalene ring, all delivered excellent enantioselectivities for both hydroarylation reactions. Electron-withdrawing substituents, such as F (**1j-k**), Cl (**1l-m**) and CF$_3$ (**1n-o**), also worked well. Given the steric requirements of ligands, we further surveyed reactants with increasingly bulky backbones. Phosphines bearing 1,2-dihydroacenaphthyl (**1p**), phenanthrenyl (**1q**), pyrenyl (**1r**) and a 1,2'-binaphthalenyl (**1s**) motif did not noticeably affect the reaction efficiency and enantioselectivity for either set of reaction conditions. We were pleased to find that heteroaryl-based phosphines, such as quinolone (**1t**) and dibenzo[*b,d*]furan (**1u**) were very well tolerated in enantioselective olefination. The reaction of alkyne **2a** with *N*-

**Table 1 | Optimization of the reaction conditions.[a]**

| Entry | Cat [M] (mol%) | L* (mol%) | Solvent | T (°C) | Ee of 3aa (%)[b] | Yield of 3aa (%)[c] |
|---|---|---|---|---|---|---|
| 1 | [Ir(cod)Cl]₂ (5) | L1 (11) | Toluene | 70 | 0 | 33 |
| 2 | [Ir(cod)Cl]₂ (5) | L2 (11) | Toluene | 70 | 51 | 79 |
| 3 | [Ir(cod)Cl]₂ (5) | L3 (11) | Toluene | 70 | 97 | 82 |
| 4 | [Ir(cod)Cl]₂ (5) | L4 (11) | Toluene | 70 | −66 | 76 |
| 5 | [Ir(cod)Cl]₂ (5) | L5 (11) | Toluene | 70 | – | <5 |
| 6 | [Ir(cod)Cl]₂ (5) | L3 (11) | THF | 70 | 71 | 44 |
| 7 | [Ir(cod)Cl]₂ (5) | L3 (11) | DCM | 70 | 83 | 23 |
| 8 | [Ir(cod)Cl]₂ (5) | L3 (11) | Toluene | 90 | 93 | 88 |
| 9 | [Ir(cod)Cl]₂ (5) | L3 (11) | Toluene | rt | 99 | 11 |
| 10 | [Ir(coe)₂Cl]₂ (5) | L3 (11) | Toluene | 70 | 77 | 77 |
| 11 | [Rh(cod)Cl]₂ (5) | L3 (11) | Toluene | 70 | 12 | 70 |
| 12 | Pd(OAc)₂ (5) | L3 (11) | Toluene | 70 | – | 0 |
| 13 | [Ir(cod)Cl]₂ (2.5) | L3 (5.5) | Toluene | 70 | 98 | 15 |
| 14 | – | L3 (11) | Toluene | 70 | – | 0 |

THF tetrahydrofuran, DCM dichloromethane.
[a]Reaction conditions: cat [M] (2.5–5 mol%), L* (5.5–11 mol%), 1a (0.2 mmol), 2a (1.0 mmol) in 2 mL of dry solvent at 70 °C for 72 h under argon.
[b]Enantiomeric excess (ee) was determined by chiral HPLC analysis.
[c]Isolated yield.

aryl pyrrole-based phosphine **1v** delivered the desired product **3va** in 69% yield and 96% ee, showing the same absolute configuration as the biaryl substrates. In addition, variations in the aryl substituents (**1w**) at the phosphorus atom with different electronic properties, were well tolerated under reaction conditions. Furthermore, a thiophenl-containing ligand **1x** also proved to be highly efficient. We then explored the scope for alkynes **2** with the developed conditions. Increasing the steric hindrance in the internal alkynes (**2b-d**) with ethyl to isobutyl groups maintained the high enantioselectivities but resulted in a gradual decrease in the reaction yield. Nonsymmetric alkynes, such as prop-1-yn-1-ylbenzene (**2e**), led to enantioenriched product **3ae** in a highly regioselective manner, albeit with diminished yield. In addition, the employment of a terminal alkyne **2f** as a substrate completely failed to undergo C−H activation in the current system.

Hydroarylation of biaryl phosphines with olefins was then studied in the transformation (Fig. 3). Initially, the different ligands and reaction parameters were evaluated with phosphine **1a** and olefin **4a** (Fig. 3a). We found that the selection of the BINOL-derived phosphoramidite **L4** under 150 °C for 48 h could give the desired product **5aa** (with ~44.8 kcal/mol rotational barrier) in 77% yield and 91% ee. The absolute configuration of this product was confirmed by X-ray diffraction, and the stereochemistry of other products was assigned by analogy to this crystal. Using the same substrates as in Fig. 2, we

next evaluated the reaction efficiency with alkene **4a** (Fig. 3b). A wide range of biaryl phosphines that incorporate electron-neutral (**1b-e**), electron-donating (**1f-i**) and electron-withdrawing (**1j-o**) substituents, were readily tolerated with alkene **4a** to produce the related products **5ba-oa** in modest to good yields and with excellent levels of enantioselectivities. Moreover, the use of polycyclic phosphine **1p-s** and heterocycle-containing phosphines **1t-u** did not interfere with productive atroposelective hydroarylation. However, the pyrrole-based phosphine **1v** only delivered the desired product **5va** in trace amounts at the current reaction conditions. Phosphines **1w-x** with different substituents at the P atom were also compatible with this reaction. In addition to alkene **4a**, we found that steric hindrance of the acrylate (**4b-d**) had a minimal impact on the reaction outcome, but styrene (**4e**) was not compatible with this C−H activation process. Finally, one example highlights the kinetic resolution of racemic 1,1′-binaphthalenyl phosphine **1y** by asymmetric hydroarylation of alkene **4a** (Fig. 3c). Chiral product **5ya** was formed in 93% ee, and the remaining starting material **1y** exhibited an excellent selectivity factor (S = 99).

**Synthetic applications**

To emphasize the practicality of our chemistry, we chose some chiral products, including **3aa**, **5aa** and **5ya**, with different steric and

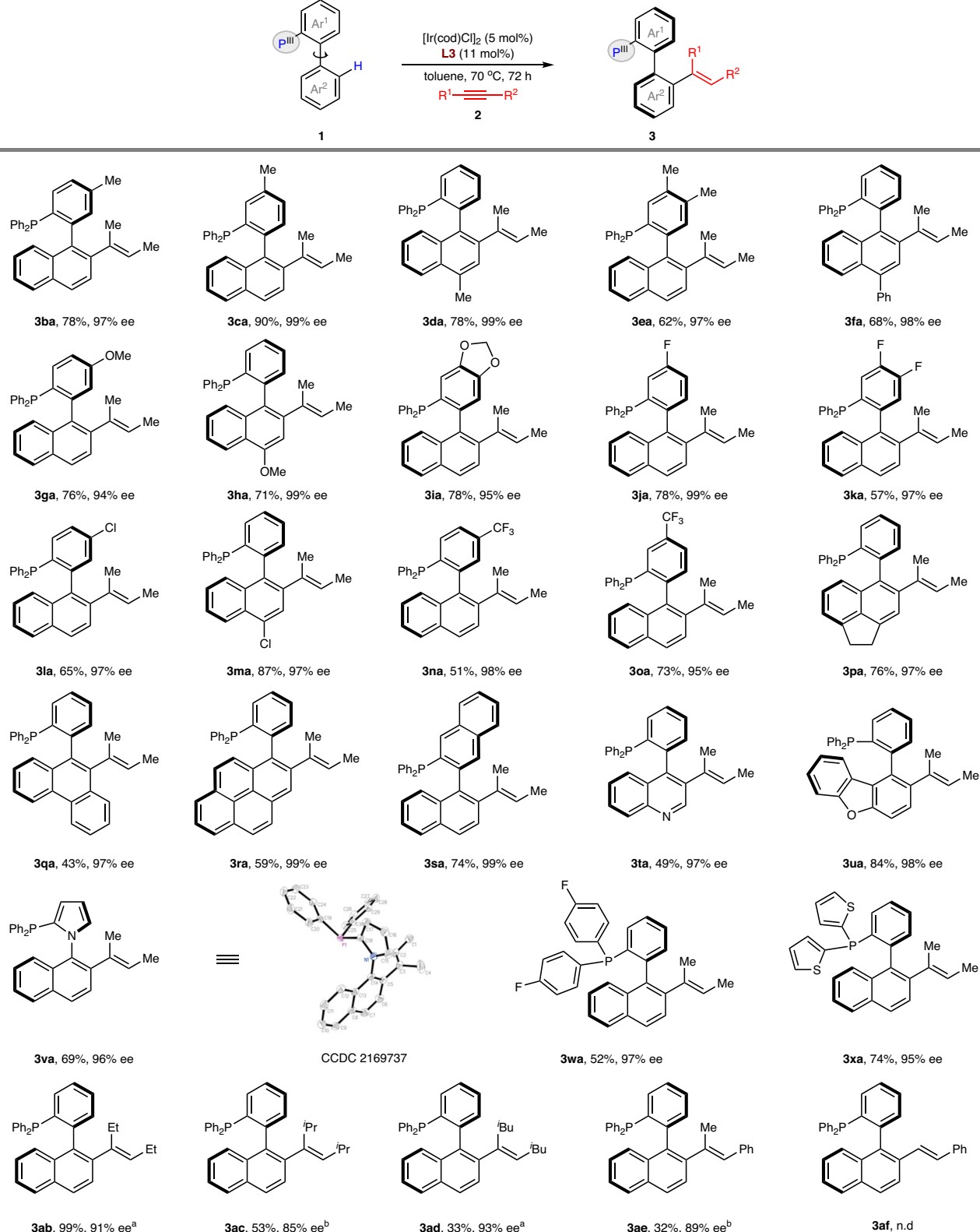

**Fig. 2 | Atroposelective P(III)-directed C–H olefination of biaryl phosphines with alkynes.** Reaction conditions: [Ir(cod)Cl]₂ (5 mol%), **L3** (11 mol%), **1** (0.2 mmol), **2** (1.0 mmol) in 2 mL of dry toluene at 70 °C for 72 h under argon; ee was determined by chiral HPLC analysis; Isolated yield. [a]At 75 °C. [b]At 90 °C.

electronic properties, to generate a ligand library and tested them for asymmetric catalysis. In the first example (Fig. 4a), use of **3aa** as a ligand in the rhodium-catalyzed arylation of isatin **6** with boronic acid **7** afforded alcohol **8** with good yield and high ee (91%, 93% ee); these were much higher than those obtained using the original MeO-MOP ligand (72%, 75% ee)[63]. In the second case (Fig. 4b), compound **5aa** showed high reactivity for palladium-catalyzed asymmetric allylic alkylation[64] with substrate **9** and dimethyl malonate (**10**), which afforded the desired product **11** in 93% yield and with 92% ee. In the third example (Fig. 4c), phosphine **5ya** was used as a ligand for

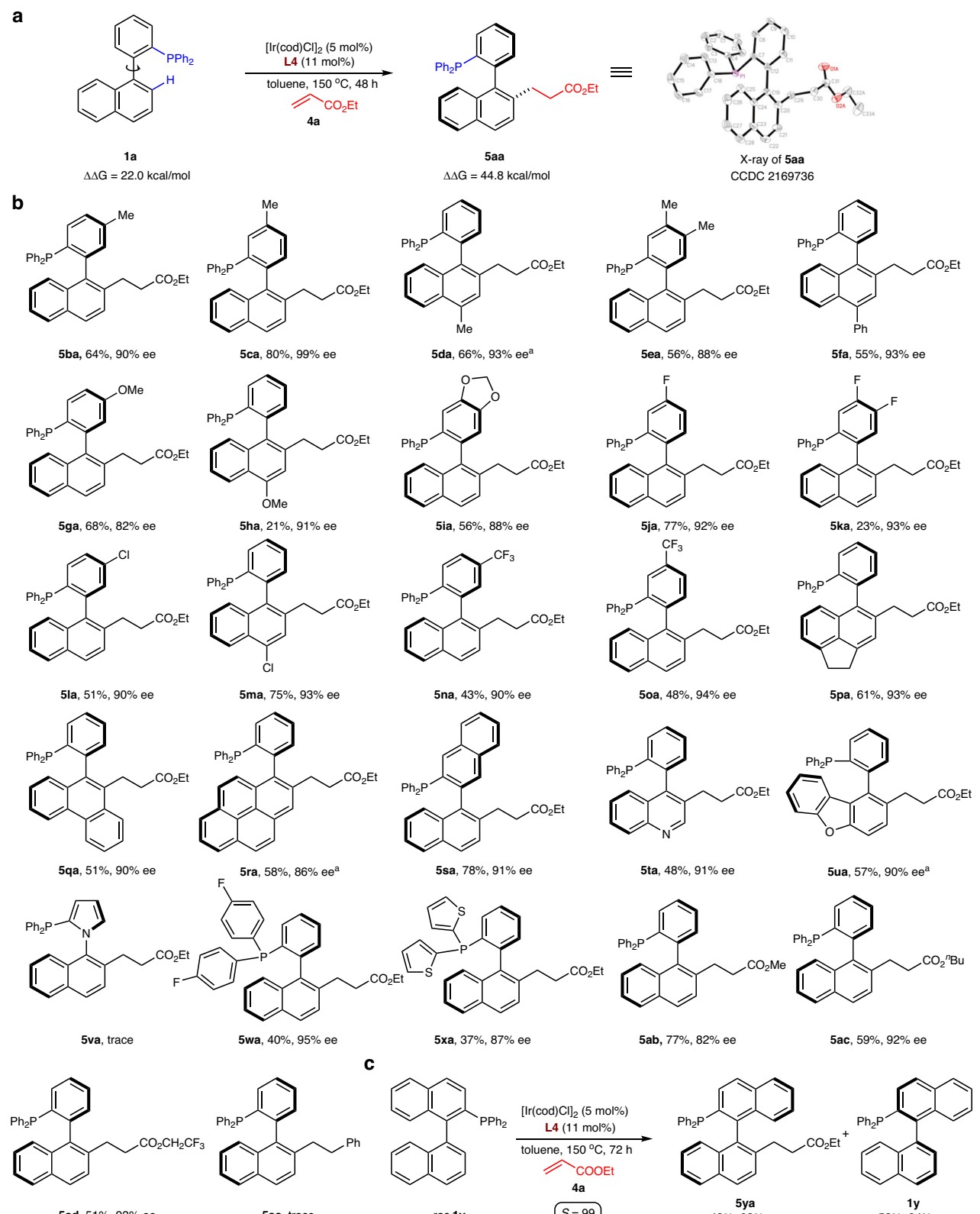

**Fig. 3 | Atroposelective P(III)-directed C−H alkylation of biaryl phosphines with alkenes. a** Atroposelective hydroarylation of phosphine **1a** with olefin **4a**. **b** Reaction substrate scope. **c**, Kinetic resolution of racemic phosphine **1y** by

palladium-catalyzed Suzuki-Miyaura cross-coupling of aryl halide **12** with boronic acid **13** to form atropisomeric biaryl **14** in 80% yield and with 93% ee[50,51]. These in situ-modified chiral ligands have proven valuable in accelerating the optimization of asymmetric catalysis.

hydroarylation of olefin **4a**. Reaction conditions: [Ir(cod)Cl]₂ (5 mol%), **L4** (11 mol%), **1** (0.2 mmol), **4** (1.0 mmol) in 2 mL of dry toluene at 150 °C for 48 h under argon; ee was determined by chiral HPLC analysis; Isolated yield. [a]Using 10.0 equiv. of **4**.

## Discussion

We next performed several experiments to investigate this asymmetric C−H activation. Reacting 1.0 equiv. of **L3** with a stoichiometric quantity of [Ir(cod)Cl]₂ yielded complex **15**, as confirmed by

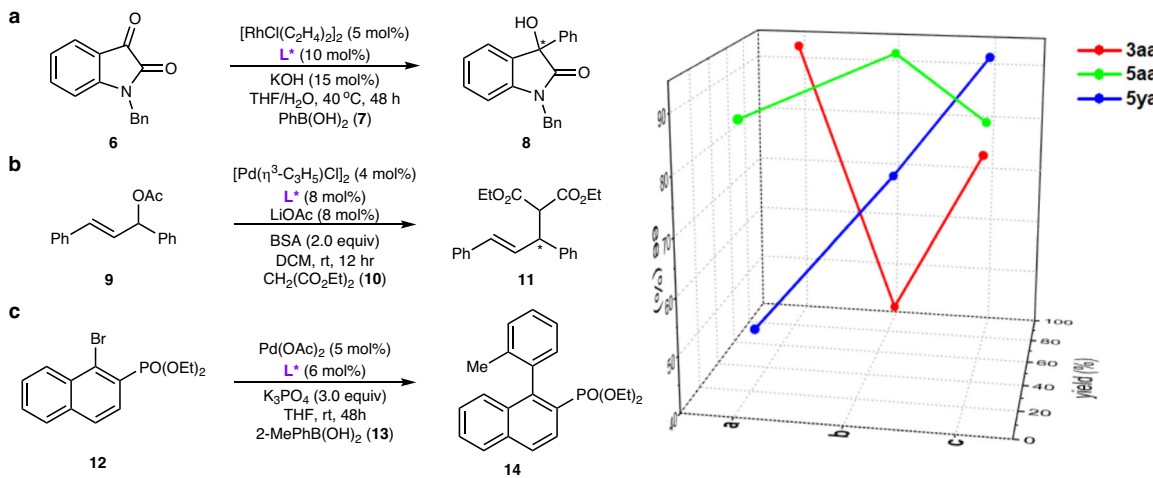

**Fig. 4 | Applications of the developed chiral phosphines in asymmetric catalysis. a** Asymmetric aryl-addition reaction. **b** Asymmetric allylic alkylation reaction. **c** Asymmetric Suzuki-Miyaura cross-coupling reaction.

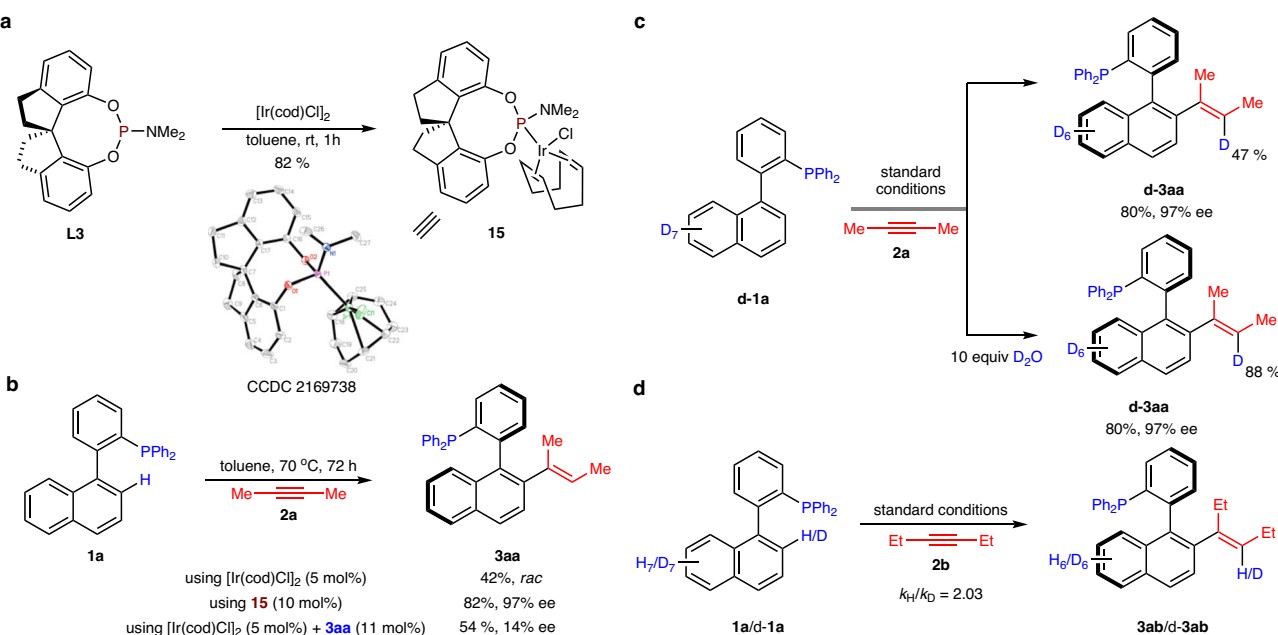

**Fig. 5 | Mechanistic experiments. a** Isolation of an iridium complex **15. b** Test of the reactivity of **15** and **3aa** for C−H activation of **1a. c** Observation of H/D exchange during the reaction of d-**1a** with alkyne **2a. d** Kinetic isotope effect (KIE) experiments.

X-ray analysis (Fig. 5a). Using **15** as the catalyst for hydroarylation of alkyne **2a** with phosphine **1a** yielded compound **3aa** with good efficiency, and addition of a catalytic quantity of **3aa** to the system as a ligand led to low enantioselectivity (Fig. 5b). These results showed that the formed products did not serve as ligands. Furthermore, a clear linear effect for the reaction of **1a** and **2b** indicated that only one chiral ligand could coordinate to the Ir center (see the Supplementary Information for details). A series of deuterium labeling experiments were then performed (Fig. 5c). The reaction of phosphine d-**1a** with alkyne **2a** resulted in 47% deuterium incorporation at the olefinic position of product **3aa**. Further addition of 10 equiv. of $D_2O$ to the system dramatically improved the level of deuterium incorporation (85% D), suggesting considerable H/D exchange occurred between the trace water in the solvent and the reaction intermediates. In addition, KIE experiments ($k_H/k_D = 2.03$) indicated that C−H bond cleavage was the rate-determining step in the reaction (Fig. 5d).[65]

The asymmetric C−H activation involves a key intermediate **INT1-L3** in which the Ir center coordinates with two phosphorus atoms in the ligand and substrate (Ir/**L3**/**1a** = 1/1/1). The high equilibrium population of this species can be illustrated by analysis of the frontier molecular orbitals (Fig. 6). The main contribution to the unoccupied molecular orbital (LUMO+2) of the intermediate **Ir-monomer** came from the Ir 5d orbital, which serves as an electronic acceptor for the phosphorus lone pair. The energy difference between **Ir-monomer** and the phosphorus-occupied molecular orbital (HOMO-1) of **1a** was 1.9 eV less than that of the LUMO+2-HOMO-4 gap between **Ir-monomer** and the chiral ligand **L3** (10.6 eV *vs.* 12.5 eV), suggesting that strong σ donation makes the electron-rich phosphine **1a** more likely to coordinate with **Ir-monomer**. Then, the η⁴-cod ligand dissociates to provide vacancies around the Ir center (**INT1-L3-pre**) to facilitate the coordination of chiral ligand. The Ir 5d occupied molecular orbital (LUMO) of **INT1-L3-pre** is thermodynamically favorable to interact with the phosphine 3d unoccupied molecular orbital (HOMO)

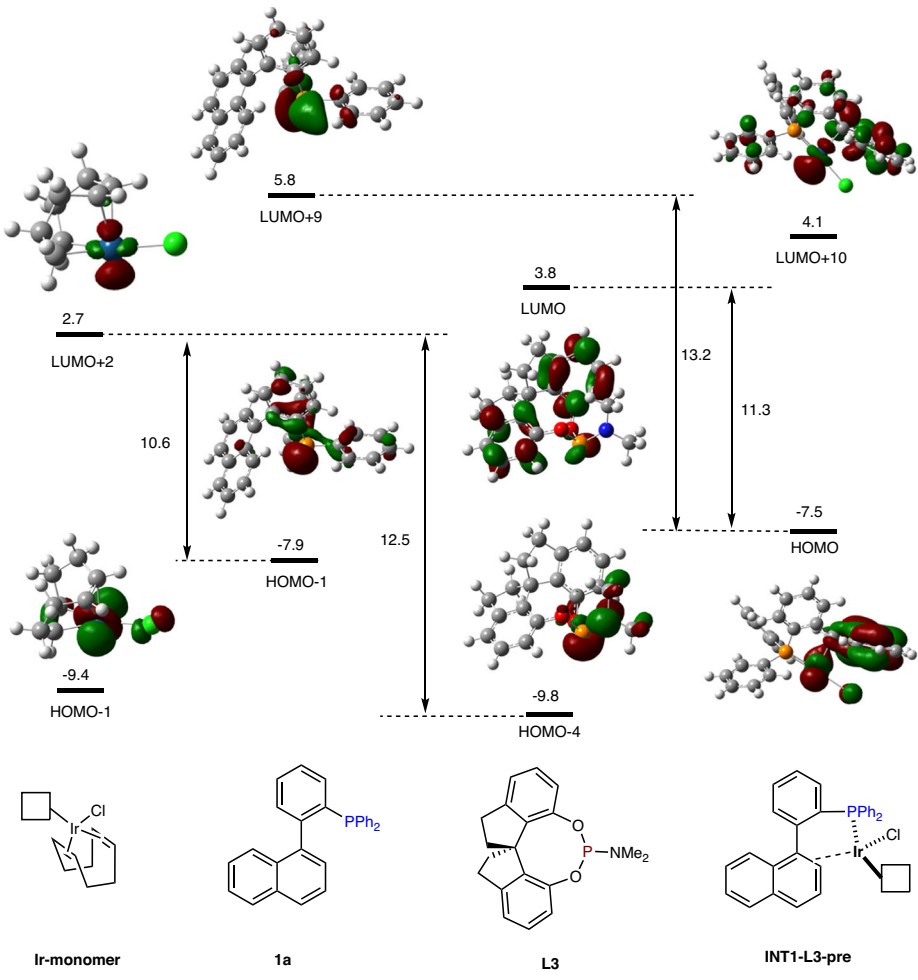

**Fig. 6 | Analysis of frontier molecular orbitals.** HF/6-31G(d)-Lanl2DZ/CPCM.

of **L3** than with another phosphine ligand **1a** (11.3 eV *vs.* 13.2 eV). This outcome indicates that the second coordination with **L3** mainly occurs via π-back bonding, thus stabilizing the formation of intermediate **INT1-L3**.

Based on the aforementioned results, the energy profile for the reaction between phosphine **1a** and alkyne **2a** is shown in Fig. 7a[66–70]. Biaryl atropisomers could easily undergo interconversion and the axial chirality of **1a** leads to the formation of two enantiomers **INT1-L3-R** and **INT1-L3-S**, where **INT1-L3-R** was set as the relative zero point of Gibbs free energy. The two enantiomers **INT1-L3** undergo C–H activation, generating the iridium complex **INT2-L3** through a reversible oxidative addition. Subsequent reductive elimination of H–Cl via transition state **TS3-L3** reversibly generates intermediate **INT3-L3**, and this process is predominantly a H/D exchange with trace water in the solvent. As a competing pathway, insertion of alkyne **2a** into the Ir–H bond of **INT2-L3-R** leads to **INT4-L3-R** via **TS4-L3-R** with a relative free energy barrier of 30.7 kcal·mol⁻¹, which is comparable in energy to the transition state **TS3-L3**, but 2.5 kcal·mol⁻¹ lower than that for the formation of **INT4-L3-S** (30.7 vs 33.2 kcal·mol⁻¹, Fig. 7b). The irreversible alkyne insertion has the highest activation energy in the catalytic cycle and is therefore proposed to be the rate-determining and enantio-determining step, in accordance with the results of KIE experiments. The stereochemical model can be further visualized by steric maps around the Ir catalyst using SambVca 2.1 tool (Fig. 7c). The geometries of **TS4-L3-R** and **TS4-L3-S** are octahedron, where Ir–Cl bond is defined as Z axis and chiral ligand **L3** is located in SE quadrant of the steric map. In both transition state **TS4-L3**s, C–H bond adjacent to the Ir–C bond towards to the

direction of Ir–Cl bond and interacts with nitrogen atom of **L3** by weak hydrogen bond, resulting **L3** appears vertically more extended in favored transition state **TS4-L3-R** and horizontally more extended in disfavored **TS4-L3-S**. The horizontal extension increases the repulsion interaction between methyl group of alkyne and phenyl ring of **L3**, resulting the energy barrier of **TS4-L3-S** is significantly higher than that of **TS4-L3-R**. The dominant intermediate **INT4-L3-R** further undergoes reductive elimination through transition state **TS5-L3-R** with an activation free energy of 13.8 kcal·mol⁻¹ to form the **INT5A-L3-R** complex. Finally, ligand exchange between **INT5A-L3-R** and phosphine **1a** releases product **3aa** and regenerate **INT1-L3** to complete the catalytic cycle.

In summary, we have demonstrated an enantioselective catalytic strategy for phosphorus-directed C–H activation enabled by chiral phosphorus ligands with iridium catalysts. This method represents an effective approach for modular syntheses of chiral phosphorus ligands via one-step transformations with widely available parent ligands, which considerably expands the toolbox of reactions available to synthetic chemists. Furthermore, the results of this study constitute a proof of principle for asymmetric C–H activations in more general molecules, which may furnish solutions for the remaining limitations in this field.

## Methods

Due to slight variations in the experimental protocols for the processes presented herein, we refer the reader to the Supplementary Information for experimental details.

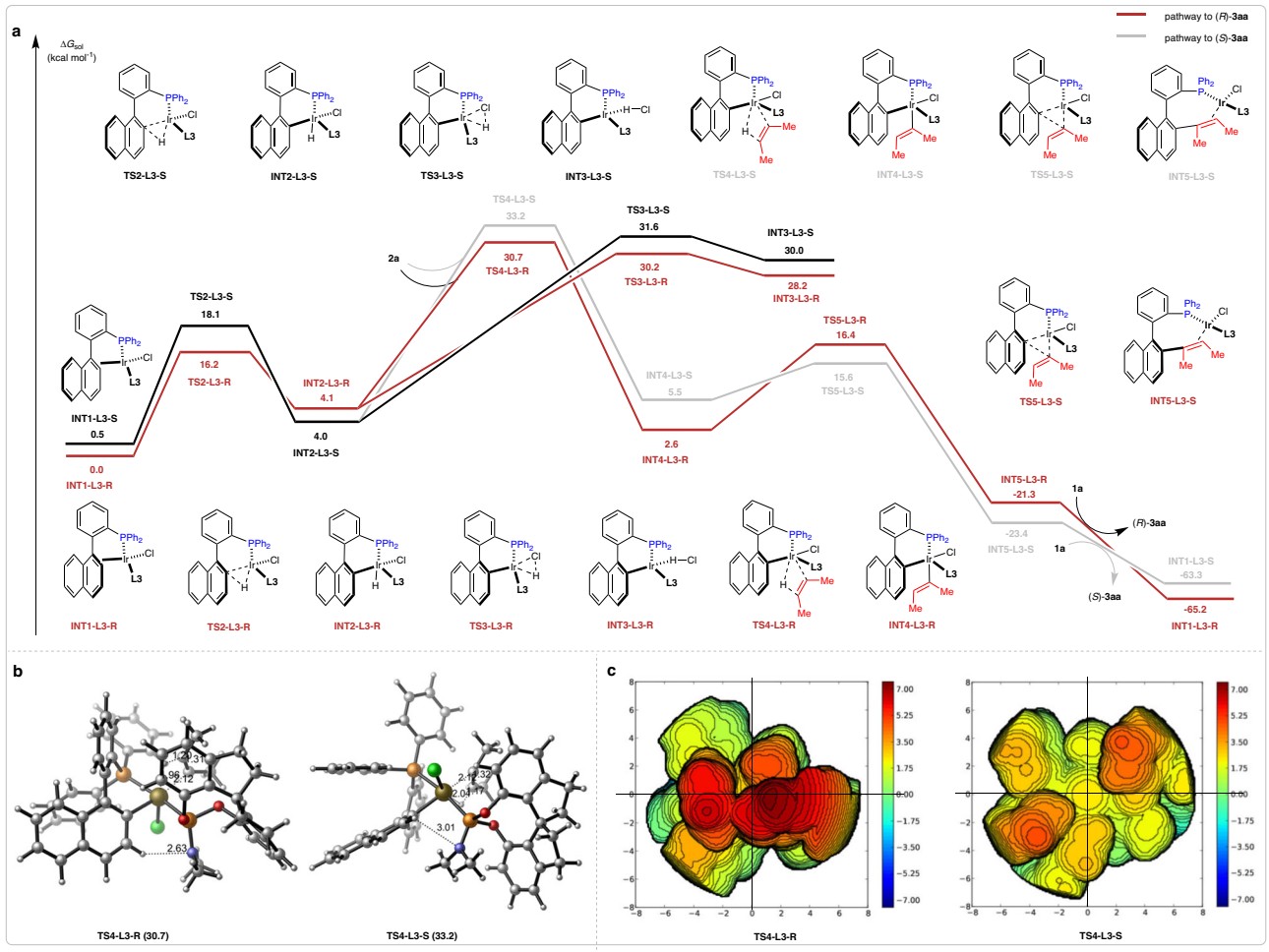

**Fig. 7 | Computational investigations. a** DFT-computed reaction pathways (ωb97xd/6-311+G(d,p)-SDD(Ir)/CPCM//B3LYP-D3BJ/6-31G(d)-Lanl2DZ(Ir)/CPCM).
**b** Stereochemical mode. **c** Steric maps of the Ir(III) transition states **TS4-L3-R** and **TS4-L3-S**.

## Data availability

Crystallographic data for the structures of **3aa**, **3va**, **5aa** and **15** reported in this paper have been deposited at the Cambridge Crystallographic Data Center under deposition numbers CCDC 2169735, 2169736, 2169737 and 2169738. Copies of the data can be obtained free of charge via www.ccdc.cam.ac.uk/getstructures. All other data supporting the findings of the study, including experimental procedures and compound characterization, are available within the paper and its Supplementary Information, or from the corresponding author upon request. Coordinates of the optimized structures are provided in the source data file. Source data are provided with this paper.

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

## Acknowledgements

We would like to thank the National Natural Science Foundation of China (Grants 22025104, 22171134 and 21972064), Natural Science Foundation of Jiangsu Province (BK20220033), Natural Science Foundation for Excellent Youth of Anhui Education Department (Grant 2022AH030056), and the Fundamental Research Funds for the Central Universities (Grant 020514380254) for their financial support. We are also grateful to the High-Performance Computing Center of Nanjing University for performing the numerical calculations in this paper on its blade cluster system.

## Author contributions

Z.S. conceived and designed the study and wrote the paper. Z.L. performed the experiments, mechanistic studies, and analyzed the data. Y.Y. made contributions during the revision. K.H and Y.L. supervised the DFT calculation and commented on the manuscript. M.W. and X.C. performed the density functional theory (DFT) calculations. Y.Z. performed the crystallographic studies.

## Competing interests

The authors declare no competing interests.
