## [Peer Review File · Nature Communications]

Atroposelective hydroarylation of biaryl phosphines directed by phosphorus centresReviewers' Comments:

Reviewer #1:

Remarks to the Author:

This article from Shi and co-workers describes an enantioselective C–H activation directed by phosphorus centers. The field of directed C–H activation is well studied, most commonly using O and N-containing directing groups. Phosphorus coordinates strongly with transition metals and is therefore challenging to use as a director in catalytic C–H activation. A number of papers have been published in the last five years using P(III) directing group in C–H activation (see refs 29-43 in the manuscript). Enantioselective control is an 'obvious' but non-trivial next step. To my knowledge, this submission is this first of its kind, enabling rapid construction of axially chiral phosphines through dynamic kinetic resolution. The demonstration of the origin of atroposelectivity has been conducted by detailed mechanistic investigation. Also noteworthy is that the in situ-modified chiral ligands have proven valuable in accelerating the optimization of asymmetric catalysis. In general, this manuscript is well-written and clear, the science is interesting and I consider it to be high-impact based on a substantial leap over the reported methods to access enantioenriched phosphorus ligands. Accordingly, I am in favor of publication in Nature Communications upon a couple of issues that need to be addressed.

1) In Figure 1b and c, the atroposelectivity of the structures didn't express clearly. The authors should redraw the related structures.

2) E.g. are all the %ees shown in Table 1 favouring the illustrated enantiomer? Given the differing nature of L1-L5, one would think that the other enantiomer would be favoured in some cases.

3) Based on the DFT calculation, the insertion of alkyne into Ir-H bond requires an activation energy barrier of 30.7 kcal mol⁻¹. A competitive pathway involving insertion of alkyne into Ir-C bond needs to be considered.

4) In the supporting information, the authors should provide the information that how the racemic product was formed and evaluated for chiral HPLC analysis in the SI.

Reviewer #2:

Remarks to the Author:

This manuscript reports on the development of asymmetric C-H activation directed by a phosphorous centre. C-H activation is a critical step in a wide-range of synthetic chemistry and while a variety of different methods to selectively activate these bonds, the addition of a new approach to this toolkit is certainly welcome. As such, I believe that this work will be of interest to a wide-range of chemists.

As a computational chemist, I have predominantly assessed the quality of the calculations and the inferences drawn from these. The choice of the calculation scheme is traditionally acceptable but not particularly modern. Why is a split optimisation/single point scheme used? The size of the systems studied are not so large that an appropriately large basis set and density functional can't be employed in their optimisation. The authors have quoted in the manuscript the method as (ωb97xd/6-311+G(d,p)-SMD/CPCM//B3LYP-D3BJ/6-31G(d)LANL2DZ/CPCM). However, this is not particularly clear. It seems to suggest that the 6-311+G(d,p) basis set was used for Ir in the calculations, which does not exist. In the supporting information this method is quoted as ωb97xd/6-311+G(d,p)[16]-SDD[17]. Presumably then the manuscript has a typo and -SMD should be -SDD, but again this should be clarified. In addition, why is a different pseudo potential basis set combination used for Ir when going from the optimisation to the single point method? Again, the SI does not help clarify this as in the SI the pseudo potential is listed as LANL4DZ[13]? I assume this is a typo, but it is not clear.

Overall, I think the the computational results support the proposed mechanism, which is encouraging. However, the difference in energy between the pathways leading to the R/S products is small (2.5 kcal/mol) and given the approximations in the approach used, I would not be confident to say that a change in method would not result in a change in outcome. The authors should repeat the calculations at a single consistent level of theory that adequately captures the chemistry during the optimisation as

well as providing the final energies from these structures.

Finally, the provision of coordinates in a PDF for the calculated structures is no longer acceptable. These structures should be uploaded to a repository to allow researchers to download these in a useable format. Based on this and the current confusion over the methods used as described in the manuscript/SI, I would have to say that there is not sufficient detail provided for the work to be reproduced.

Reviewer #3:

Remarks to the Author:

The manuscript by Shi and co-workers reported Iridium catalyzed preparation of atropisomeric biaryls through phosphine directed asymmetric C-H functionalization. This work is most likely to be the first example of phosphine directed catalytic asymmetric C-H functionalization. Compared with phosphine oxides directed asymmetric C-H functionalization that requires a follow-up reduction step, this protocol provides a more straightforward access to chiral phosphine ligands. Key to their success is the identification of a chiral spiro phosphoramidite as the chiral ligand. Both internal alkynes and acrylates are amenable to insert into the target C-H bonds in good yields and high enantioselectivities. Some of the products could act as good chiral ligand in several catalytic asymmetric reactions, highlighting the remarkable synthetic utility of this protocol. A series of experiments and calculations were carried out to shed light on the mechanism of this reaction, suggesting that the Ir-center is coordinated with only one chiral ligand and a substrate in the active intermediate. The energy values in a number of follow-up steps are different for the two diastereoisomers of this complex, thus leading to overall enantiocontrol. I think this work is very important for the field of asymmetric C-H activation and I suggest acceptance of this work in Nat. Comm. after very minor revision.

1. The condition screening is suggested to be added in supplementary materials.

2. I'm very interested in the experiment in fig. 5b that utilized product 3aa as the chiral ligand for asymmetric insertion of the internal alkyne. Only moderate yield and low ee (14%) was observed in their results. However, I think product 3aa is much bulkier and therefore a weaker ligand than the substrate 1a. The main reason for the nearly racemic result is more likely to be the coordination of excess racemic substrates as the ligand. Therefore, an experiment without additional chiral ligand is suggested to be carried out.

Reviewer #1 (Remarks to the Author):

This article from Shi and co-workers describes an enantioselective C–H activation directed by phosphorus centers. The field of directed C–H activation is well studied, most commonly using O and N-containing directing groups. Phosphorus coordinates strongly with transition metals and is therefore challenging to use as a director in catalytic C–H activation. A number of papers have been published in the last five years using P(III) directing group in C–H activation (see refs 29-43 in the manuscript). Enantioselective control is an 'obvious' but non-trivial next step. To my knowledge, this submission is the first of its kind, enabling rapid construction of axially chiral phosphines through dynamic kinetic resolution. The demonstration of the origin of atroposelectivity has been conducted by detailed mechanistic investigation. Also noteworthy is that the in situ-modified chiral ligands have proven valuable in accelerating the optimization of asymmetric catalysis. In general, this manuscript is well-written and clear, the science is interesting and I consider it to be high-impact based on a substantial leap over the reported methods to access enantioenriched phosphorus ligands. Accordingly, I am in favor of publication in Nature Communications upon a couple of issues that need to be addressed.

Response: Thanks for the kind comments.

1) In Figure 1b and c, the atroposelectivity of the structures didn't express clearly. The authors should redraw the related structures.

Response: We revised the related structures.

E.g. are all the %ees shown in Table 1 favouring the illustrated enantiomer? Given the differing nature of L1-L5, one would think that the other enantiomer would be favoured in some cases.

Response: Other enantiomer would be favored in some ligands. We modified the Table according to your suggestion.

2) Based on the DFT calculation, the insertion of alkyne into Ir-H bond requires an activation energy barrier of 30.7 kcal mol⁻¹. A competitive pathway involving

insertion of alkyne into Ir-C bond needs to be considered.

Response: Thanks for constructive suggestions. DFT calculation was also performed to calculate the energy barriers of the competitive pathways for the insertion of alkyne into Ir-H and Ir-C bond of intermediate **INT2-L3-R**, respectively. The alkyne is more prone to insert into Ir-H bond of intermediate **INT2-L3-R** through four-membered cyclic transition state **TS4-L3-R** with a free energy of 26.7 kcal·mol⁻¹ (30.7 kcal·mol⁻¹ relative to **ITN1-L3-R**), which is much lower than the insertion pathway into Ir-C bond via transition state **TS4'-L3-R** (26.7 vs 39.9 kcal·mol⁻¹). The relevant results have been added in the revised Supporting Information.

3) In the supporting information, the authors should provide the information that how the racemic product was formed and evaluated for chiral HPLC analysis in the SI.

Response: We added the information according to your suggestion.

Reviewer #2 (Remarks to the Author):

This manuscript reports on the development of asymmetric C-H activation directed by a phosphorous centre. C-H activation is a critical step in a wide-range of synthetic chemistry and while as variety of different methods to selectively activate these bonds,

the addition of a new approach to this toolkit is certainly welcome. As such, I believe that this work will be of interest to a wide-range of chemists.

As a computational chemist, I have predominantly assessed the quality of the calculations and the inferences drawn from these. The choice of the calculation scheme is traditionally acceptable but not particularly modern. Why is a split optimisation/single point scheme used? The size of the systems studied are not so large that an appropriately large basis set and density functional can't be employed in their optimisation. The authors have quoted in the manuscript the method as (ω b97xd/6-311+G(d,p)-SMD/CPCM//B3LYP-D3BJ/6-31G(d)LanL2DZ/CPCM).

However, this is not particularly clear. It seems to suggest that the 6-311+G(d,p) basis set was used for Ir in the calculations, which does not exist. In the supporting information this method is quoted as ω b97xd/6-311+G(d,p)[16]-SDD[17]. Presumably then the manuscript has a typo and -SMD should be -SDD, but again this should be clarified. In addition, why is a different pseudo potential basis set combination used for Ir when going from the optimisation to the single point method? Again, the SI does not help clarify this as in the SI the pseudo potential is listed as LANL4DZ[13]? I assume this is a typo, but it is not clear.

Response: Thanks for the constructive suggestions. The former mistakes have been checked and revised. In our manuscript, B3LYP-D3BJ method together with LANL2DZ basis set for iridium atom and 6-31G(d) basis set for all other atoms was used to optimize all geometries in toluene solvent with CPCM solvation model. Frequency computations were carried out at the same level to identify all of the stationary points and to provide the thermal correction to free energies.

In addition, ω b97x-D functional with larger basis set 6-311+G (d,p)-SDD(Ir) was chosen to optimize the geometries and calculate the frequencies. The calculated energies of the key transition states and intermediate were listed in the following Table:

Intermediate	$G_{\text{b3lyp-d3bj}} + E_{\omega\text{b97xd}}$ (kcal mol ⁻¹)	$G_{\omega\text{b97x-D/6-311+G (d,p)-SDD/CPCM}}$
--------------	--	--

 INT1-L3-R	0.0	0.0
 TS2-L3-R	16.2	15.5
 INT2-L3-R	4.1	3.1
 INT1-L3-S	0.5	0.4
 TS2-L3-S	18.1	16.8
 INT2-L3-S	4.0	3.3

The results indicated the relative energy barriers are almost consistent with the energy barriers calculated by the method we used in the supporting information. The larger basis set 6-311+G(d,p)-SDD(Ir) needs to take longer calculation times. Further considering the computational efficiency and cost, the suitable basis set 6-31(G)-LANL2DZ(Ir) were chosen to optimize the geometry of intermediates and transition states.

Overall, I think the computational results support the proposed mechanism, which is encouraging. However, the difference in energy between the pathways leading to the R/S products is small (2.5 kcal/mol) and given the approximations in the approach used, I would not be confident to say that a change in method would not result in a change in outcome. The authors should repeat the calculations at a single consistent level of theory that adequately captures the chemistry during the optimisation as well as providing the final energies from these structures.

Response: Thanks for the constructive suggestion. Three DFT functionals (B3LYP-D3BJ, M06, and ω b97x-D) with SDD for Ir and 6-311+G(d,p) for other atoms was further used to compute the solvation single-point energies in toluene with CPCM continuum model. All of the energies are calculated as the sum of the solution-phase free energy and the corresponding thermal correction obtained in gas phase. The absolute (in Hartree) single-point energies and relative (in kcal mol⁻¹) Gibbs free energies in toluene with different DFT functionals were shown as follows:

structure	G _{corr}	E _{B3LYP-D3BJ}	Δ G _{B3LYP-D3BJ}	E _{M06}	Δ G _{M06}	E _{ωb97xd}	Δ G _{ωb97xd}
INT1A-R	0.684646	-3269.263476	0.0	-3267.321030	0.0	-3268.281437	0.0
INT1A-S	0.685821	-3269.264195	-0.1	-3267.320236	0.8	-3268.281758	0.5
TS2A-R	0.677794	-3269.231246	15.9	-3267.288953	15.8	-3268.248752	16.2
INT2A-R	0.678548	-3269.252669	3.0	-3267.300199	9.2	-3268.268841	4.1
TS2A-S	0.680096	-3269.232837	16.0	-3267.285379	19.1	-3268.247977	18.1
INT2A-S	0.680642	-3269.256038	1.8	-3267.301772	9.2	-3268.271031	4.0
TS3A-R	0.675373	-3269.212478	26.2	-3267.257176	34.3	-3268.224113	30.2
INT3A-R	0.675567	-3269.216703	23.7	-3267.262091	31.3	-3268.227490	28.2
TS3A-S	0.677579	-3269.212429	27.2	-3267.257480	35.0	-3268.224022	31.6
INT3A-S	0.680376	-3269.217688	25.7	-3267.261057	34.6	-3268.227128	31.4
TS4A-R	0.762188	-3425.273030	30.1	-3423.172342	38.0	-3424.216769	30.7
INT4A-R	0.767390	-3425.325659	0.3	-3423.220487	11.0	-3424.266823	2.6
TS4A-S	0.762890	-3425.270138	32.0	-3423.168350	40.5	-3424.213510	33.2
INT4A-S	0.770893	-3425.326087	1.9	-3423.219206	13.6	-3424.265716	5.5

TS5A-R	0.767152	-3425.303832	13.9	-3423.205327	20.4	-3424.244645	16.4
INT5A-R	0.773053	-3425.364097	-20.2	-3423.266876	-14.5	-3424.310609	-21.3
TS5A-S	0.768067	-3425.305750	12.9	-3423.206947	19.6	-3424.246827	15.6
INT5A-S	0.774088	-3425.367967	-22.4	-3423.270868	-16.8	-3424.314939	-23.4

The calculated results indicated the solvation single-point energies of key transition states using ω b97x-D functional is well agreement with the experimental observed enantioselectivity. We have added this table in the revised Supporting Information.

Finally, the provision of coordinates in a PDF for the calculated structures is no longer acceptable. These structures should be uploaded to a repository to allow researchers to download these in a useable format. Based on this and the current confusion over the methods used as described in the manuscript/SI, I would have to say that there is not sufficient detail provided for the work to be reproduced.

Response: Thanks for the constructive suggestions. In order to allow researchers to reproduce our calculations well, we added detailed input files and output information for all of the structures to improve reproducibility and accessibility.

Reviewer #3 (Remarks to the Author):

The manuscript by Shi and co-workers reported Iridium catalyzed preparation of atropisomeric biaryls through phosphine directed asymmetric C-H functionalization. This work is most likely to be the first example of phosphine directed catalytic asymmetric C-H functionalization. Compared with phosphine oxides directed asymmetric C-H functionalization that requires a follow-up reduction step, this protocol provides a more straightforward access to chiral phosphine ligands. Key to their success is the identification of a chiral spiro phosphoramidite as the chiral ligand. Both internal alkynes and acrylates are amenable to insert into the target C-H bonds in good yields and high enantioselectivities. Some of the products could act as good chiral ligand in several catalytic asymmetric reactions, highlighting the remarkable synthetic utility of this protocol. A series of experiments and calculations were carried

out to shed light on the mechanism of this reaction, suggesting that the Ir-center is coordinated with only one chiral ligand and a substrate in the active intermediate. The energy values in a number of follow-up steps are different for the two diastereoisomers of this complex, thus leading to overall enantiocontrol. I think this work is very important for the field of asymmetric C-H activation and I suggest acceptance of this work in Nat. Commun. after very minor revision.

Response: Thanks for the kind comments.

1. The condition screening is suggested to be added in supplementary materials.

Response: We added the condition screening in the SI according to your suggestion.

2. I'm very interested in the experiment in fig. 5b that utilized product **3aa** as the chiral ligand for asymmetric insertion of the internal alkyne. Only moderate yield and low ee (14%) was observed in their results. However, I think product **3aa** is much bulkier and therefore a weaker ligand than the substrate **1a**. The main reason for the nearly racemic result is more likely to be the coordination of excess racemic substrates as the ligand. Therefore, an experiment without additional chiral ligand is suggested to be carried out.

Response: Thank you for the constructive suggestion. We conducted an additional experiment without an additional chiral ligand, as suggested, and obtained the same result as you had expected.

Reviewers' Comments:

Reviewer #1:

Remarks to the Author:

In this revised manuscript, the authors have addressed my previous comments in full. Therefore, acceptance of the revised manuscript is recommended.

Reviewer #2:

Remarks to the Author:

The authors have addressed my concerns from the first version of the manuscript and the additional information provided in the supporting information regarding the calculated values with different functionals in appropriate.

Reviewer #3:

Remarks to the Author:

The authors have properly revised the manuscript and addressed the concerns raised by the reviewers. The publication of this nice work in Nat. Commun. is highly recommended.

Reviewer #1 (Remarks to the Author):

In this revised manuscript, the authors have addressed my previous comments in full. Therefore, acceptance of the revised manuscript is recommended.

Response: Thanks for the kind comments.

Reviewer #2 (Remarks to the Author):

The authors have addressed my concerns from the first version of the manuscript and the additional information provided in the supporting information regarding the calculated values with different functionals in appropriate.

Response: Thanks for the kind comments.

Reviewer #3 (Remarks to the Author):

The authors have properly revised the manuscript and addressed the concerns raised by the reviewers. The publication of this nice work in Nat. Commun. is highly recommended.

Response: Thanks for the kind comments.